

# Mining of candidate genes involved in the biosynthesis of dextrorotatory borneol in *Cinnamomum burmannii* by transcriptomic analysis on three chemotypes

Zerui Yang, Wenli An, Shanshan Liu, Yuying Huang, Chunzhu Xie, Song Huang and Xiasheng Zheng

National Engineering Research Center for Modernization of Traditional Chinese Medicine, Mathematical Engineering Academy of Chinese Medicine, Guangzhou University of Chinese Medicine, Guangzhou, Guangdong, China

Corresponding authors
Song Huang, huangnn421@163.com
Xiasheng Zheng,
xszheng@gzucm.edu.cn,
zxs2008_gz@hotmail.com

## ABSTRACT

**Background**. Dextrorotatory borneol (D-borneol), a cyclic monoterpene, is widely used in traditional Chinese medicine as an efficient topical analgesic drug. Fresh leaves of *Cinnamomum* trees, e.g., *C. burmannii* and *C. camphor*, are the main sources from which D-borneol is extracted by steam distillation, yet with low yields. Insufficient supply of D-borneol has hampered its clinical use and production of patent remedies for a long time. Biological synthesis of D-borneol offers an additional approach; however, mechanisms of D-borneol biosynthesis remain mostly unresolved. Hence, it is important and necessary to elucidate the biosynthetic pathway of D-borneol.

**Results**. Comparative analysis on the gene expression patterns of different D-borneol production *C. burmannii* samples facilitates elucidation on the underlying biosynthetic pathway of D-borneol. Herein, we collected three different chemotypes of *C. burmannii*, which harbor different contents of D-borneol. A total of 100,218 unigenes with an N50 of 1,128 bp were assembled de novo using Trinity from a total of 21.21 Gb clean bases. We used BLASTx analysis against several public databases to annotate 45,485 unigenes (45.38%) to at least one database, among which 82 unigenes were assigned to terpenoid biosynthesis pathways by KEGG annotation. In addition, we defined 8,860 unigenes as differentially expressed genes (DEGs), among which 13 DEGs were associated with terpenoid biosynthesis pathways. One 1-deoxy-D-xylulose-5-phosphate synthase (DXS) and two monoterpene synthase, designated as *CbDXS9*, *CbTPS2* and *CbTPS3*, were up-regulated in the high-borneol group compared to the low-borneol and borneol-free groups, and might be vital to biosynthesis of D-borneol in *C. burmannii*. In addition, we identified one WRKY, two BHLH, one AP2/ERF and three MYB candidate genes, which exhibited the same expression patterns as *CbTPS2* and *CbTPS3*, suggesting that these transcription factors might potentially regulate D-borneol biosynthesis. Finally, quantitative real-time PCR was conducted to detect the actual expression level of those candidate genes related to the D-borneol biosynthesis pathway, and the result showed that the expression patterns of the candidate genes related to D-borneol biosynthesis were basically consistent with those revealed by transcriptome analysis.

**Conclusions**. We used transcriptome sequencing to analyze three different chemotypes of *C. burmannii*, identifying three candidate structural genes (one DXS, two monoterpene synthases) and seven potential transcription factor candidates (one WRKY, two BHLH, one AP2/ERF and three MYB) involved in D-borneol biosynthesis. These results provide new insight into our understanding of the production and accumulation of D-borneol in *C. burmannii*.

# INTRODUCTION

D-Borneol, also known as Tianran Bingpian in Chinese, is a time-honored drug in traditional Chinese medicine used for disease prevention and treatment for more than 2,000 years (*Luo, 2015*; *Pharmacopoeia, 2015*; *Li, 2004*). D-Borneol can be administered either transdermally or orally, with the former most frequently used. Transdermal administration is mainly used to ease pain resulting from wounds, injuries, burns, cuts and similar. When administered orally, D-borneol is widely used to treat cardiovascular diseases, including stroke, coronary heart disease and angina pectoris, as an indispensable ingredient in many traditional Chinese herbal formulas, such as Angong Niuhuang pill, Suxiao Jiuxin pill and Fufang Danshen pill (or Compound Danshen) (*Chai et al., 2019*; *Chen et al., 2019*; *Huang et al., 2016*; *Liang et al., 2018*; *Ren et al., 2018*).

Natural sources of D-borneol include the resin and essential oils of several plant families, including Dipterocarpaceae, Lamiaceae, Valerianaceae and Asteraceae. Among these, the *Cinnamomum* genus within Lamiaceae contributes the majority of the sources of D-boreol. *C. burmannii* harbors D-borneol (19.68%–78.6%) as the major component of its essential oil (*Chen et al., 2011*; *Li et al., 1987*). Steam distillation of *C. burmannii* leaves is the current approach for obtaining D-borneol. However, this traditional extraction method is costly and consumes a great amount of labor and energy. Moreover, it is difficult to achieve sustainable industrial production due to limited natural resources. Synthetic borneol, which is chemically synthesized from turpentine oil or camphor, is essentially impure, containing unexpected byproducts such as levogyral borneol and optically inactive isoborneol. In addition to those byproducts, residual raw material from the chemical reaction producing borneol, e.g., camphor, appears to be harmful to the human body (*Cheng et al., 2013*; *Mathen et al., 2018*; *Nchinech et al., 2019*; *Yang et al., 2018*). Furthermore, it has been reported that natural D-borneol exhibits higher efficacy than synthetic borneol (*Zou et al., 2017*). Therefore, the application of synthetic borneol has been restricted. In light of this, it is important to provide a new and sustainable source, e.g., heterologous reconstitution of biosynthesis, for the production of D-borneol.

D-Borneol is a bicyclic monoterpene, one of the members of the terpene family. The metabolic pathways of volatile terpenes have been well characterized in the plant kingdom (*Alquezar et al., 2017*; *Bohlmann, Meyer-Gauen & Wise, 1998*). Generally, these

terpenoids are all derived from the common precursors isopentyl diphosphate (IPP) and dimethylallyl diphosphate (DMAPP). IPP is biosynthesized in plants by the mevalonate pathway in the cytoplasm or through the methylerythritol phosphate pathway in the plastid by multi-step reactions starting from acetyl-CoA or oleic acid, respectively. IPP can then be converted into DMAPP by the catalysis of isopentyl pyrophosphate isomerase (IPPI) (Stage 1, Fig. 1A). IPP and DMAPP are further catalyzed by geranyl diphosphate synthase (GPPS), geranyl geranyl diphosphate synthase (GGPPS) or farnesyl diphosphate synthase (FPPS) to form geranyl diphosphate (GPP), geranyl geranyl diphosphate (GGPP) or farnesyl diphosphate (FPP) (Stage 2, Fig. 1A). These prenyl diphosphates are subsequently catalyzed into mono-, sesqui- or diterpenes by terpene synthase (TPS) (Stage 3, Fig. 1A). The biosynthesis pathway of D-borneol has been partially elucidated in previous studies. Firstly, GPP is converted to bornyl pyrophosphase (BPP) by the catalysis of bornyl pyrophosphase synthase (BPPS). BPP is then dephosphorylated to form borneol (Fig. 1B) (*Wise et al., 1998*; *Despinasse et al., 2017*; *Hurd, Kwon & Ro, 2017*; *Wang et al., 2018*). The *BPPS* gene has been functionally characterized in four species, *Salvia officinalis*, *Lavandula angustifolia*, *Lippia dulcis* and *Amomum villosum*. Nevertheless, these four species are not rich in D-borneol, containing only 0.98%, 22.63%, 1.12% and 2.87% D-borneol in their essential oils, respectively. Therefore, BPPS enzymes isolated from these species are of low catalytic efficiency. *C. burmannii* contains much more D-borneol than any of the above-mentioned species, and thus represents an ideal material for mining potential genes related to D-borneol biosynthesis with high catalytic efficiency (*Cardia et al., 2018*; *Compadre, Robbins & Kinghorn, 1986*; *Koubaa et al., 2019*). Nevertheless, to the best of our knowledge, genes which are responsible for the D-borneol biosynthesis in *C. burmannii* still remain unknown due to the lack of genomic and transcriptomic data.

Transcriptome studies provide a useful perspective for expounding the molecular mechanisms of gene functions, cellular reactions and different biological processes (*Kashyap et al., 2020*; *Quintana-Escobar et al., 2019*). To date, many transcriptome analyses aimed at elucidating molecular mechanisms related to biosynthesis of major terpenoids have been performed in *Cinnamomum* plants, such as *Cinnamomum camphora* (*Chen et al., 2018*) and *Cinnamomum porrectum* (*Qiu et al., 2019*). Genes with specific expression or higher expression in one of several chemotypes have been identified, laying the foundation for subsequent identification of their corresponding functions. Systematic analysis of genes involved in D-borneol biosynthesis through high-throughput sequencing in *C. burmannii* will significantly aid mining of a high-efficiency enzyme source for the construction of microbially engineered bacteria with high yields of D-borneol. Here, we compared the transcriptomes of *C. burmannii* chemotypes with three different D-borneol contents by high-throughput sequencing to seek candidate genes related to D-borneol biosynthesis.

## MATERIAL AND METHODS

### Plant material, chemicals and reagents

Ten-year-old *Cinnamomum burmannii* of low-content borneol type (LBT, 0 <D-borneol <1 mg/g) and borneol-free type (FBT) were planted in the Medicinal Botanical Garden

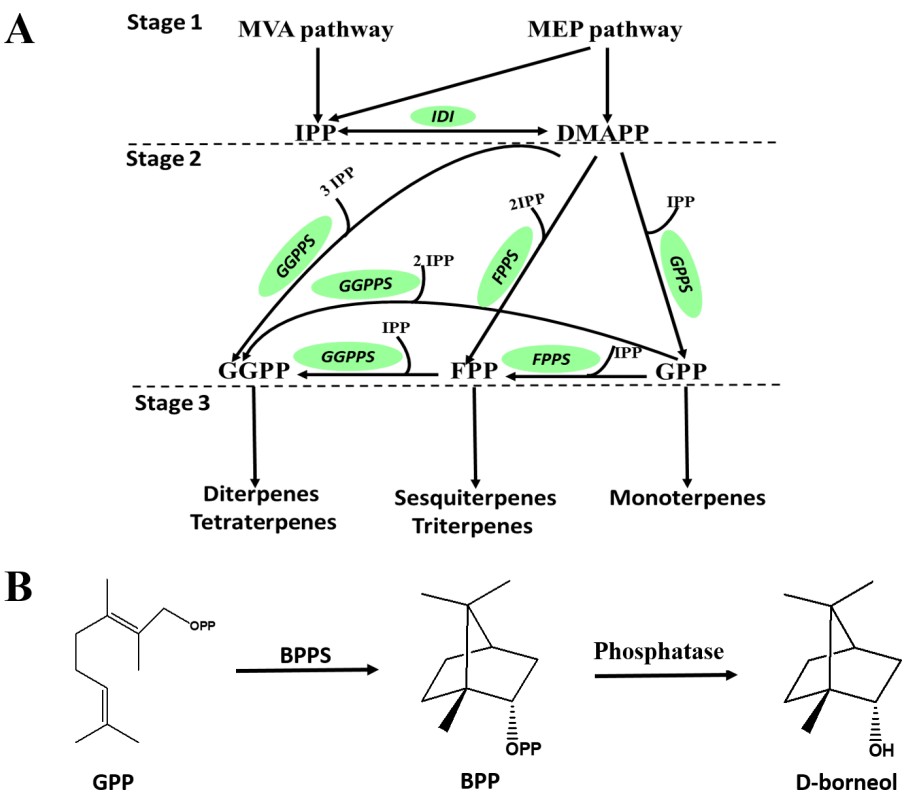

**Figure 1  Biosynthetic pathway of terpenes and biosynthetic steps from BPP to D-borneol.** Each solid arrow represents a biosynthetic reaction step. (A) Biosynthetic pathway of terpenes; (B) biosynthetic steps from BPP to D-borneol.

of Guangzhou University of Chinese Medicine. Samples of ten-year-old *C. burmannii* of high-content borneol type (HBT; D-borneol ≥ 1 mg/g) samples were donated by Guangdong Huaqingyuan Biotechnology Co. LTD. Authentic standard D-borneol (purity >98%) was purchased from Shanghai Yuanye Bio-Technology Co., Ltd.

## Gas chromatograohy-mass spectrometry (GC/MS) analysis of chemical contents

Leaves collected from different individual trees of *C. burmannii* were ground into fine powder in liquid nitrogen. Powder samples (0.05–0.10 g) were weighed accurately, soaked in 2.0 mL petroleum ether using an ultrasonic cleaner for 30 min, and centrifuged at 16,000 rpm for 5 min in preparation for GC/MS analysis. Three technical replicates were performed for each of the three biological samples. An Agilent 7890B Gas Chromatograph with 5977A Inert Mass Selective Detector (Agilent, USA) was used to analyze the extraction. Helium was used as the carrier gas with a flow rate of 1 mL/min. A Cyclosil-B GC column (30 m × 0.25 mm × 0.25 μm) with initial temperature of 50 °C was used to separate samples. The GC column temperature program was as follows: oven temperature was increased from 50 °C to 180 °C at 2 °C/min, and then from 180 °C to 300 °C at 4 °C/min, then held at 300 °C for 4 min. Injector: split mode with split ratio of 20:1, injection volume

of 1.0 µL and inlet temperature of 250 °C. D-Borneol was confirmed by comparing its retention time with that of the known standard, which was determined under the same conditions. The concentration of D-borneol in the plant sample was calculated from its respective standard curves.

## RNA extraction, cDNA library preparation and sequencing

Total RNA of samples collected from those three groups was extracted using TRizol reagent according to the manufacturer's instructions and then was digested with RNase-free DNase to eliminate the contamination of genomic DNA. The integrity and quantity of the RNA samples were analyzed using 1% agarose gel electrophoresis and with a NanoDrop 2000 Spectrophotometer (Thermo Scientific, USA). Qualified RNA samples ($OD_{260}/_{280}$ = 1.8~2.2, $OD_{260}/_{230}$ ≥ 2.0, RIN ≥ 6.5, 28S:18S ≥ 1.0) were immediately frozen at −80 °C until use. To analyze the transcriptome, 2 µg qualified RNA samples from three biological replicates were mixed and sent to Majorbio (Shanghai, China) to construct a cDNA library on an Illumina Hiseq X Ten (Illumina Inc., San Diego, CA, USA). Paired-end (PE) reads were generated and checked by the software fastx_toolkit_0.0.14 (http://hannonlab.cshl.edu/fastx_toolkit/) to assess the quality of sequences.

### *De novo* transcriptome assembly and annotation

High-quality sequencing data was *de novo* assembled using TRINITY v2.5.0 (https://github.com/trinityrnaseq/trinityrnaseq) with default parameters, which integrates three independent software modules (Inchworm, Chrysalis and Butterfly) to process and splice a large amount of RNA-Seq data in turn. First, the Inchworm software was used to assemble the clean data into linear contigs. Second, the overlapping contigs were clustered into sets of connected components using the Chrysalis software. Finally, transcripts were constructed using the Butterfly software. The assembly results were further filtered and optimized using TransRate (http://hibberdlab.com/transrate/) and clustered to obtain non-redundant unigenes using CD-HIT (http://weizhongli-lab.org/cd-hit/). For gene identification and expression analysis, the reads from different chemotypes were co-assembled, and for gene sequence analysis, reads from different chemotypes were assembled separately.

Annotation of unigenes was performed using the BLAST program with an *E*-value cut-off of 1E-5 against the following three databases: NCBI non-redundant protein sequences (Nr, https://blast.ncbi.nlm.nih.gov/), Protein family (Pfam, https://pfam.sanger.ac.uk/) and Swiss-Prot (https://web.expasy.org/docs/swiss-protguideline. html). Functional annotation using gene ontology (GO) terms was analyzed using Blast2GO version 2.5.0 (http://www.geneontology.org/). Kyoto encyclopedia of genes and genomes (KEGG) pathways analysis was performed using KOBAS version 2.1.1 (https://www.genome.jp/kegg/) for systematic analysis of gene function.

## Differential gene expression analysis

Fragments Per Kilobases per Million-reads (FPKM) was used to estimate relative expression levels of transcripts using the software RSEM v1.2.15 (http://deweylab.github.io/RSEM/) with default parameters. Raw counts were directly analyzed statistically using DESeq2 version 1.10.1 software based on the negative binomial distribution. Unigenes with

expression differences among the groups were obtained with the parameters p-adjust <0.05 and |log2FC| ≥ 1. KEGG enrichment analysis of differentially expressed genes (DEGs) was performed using KOBAS version 2.1.1 (http://www.genome.jp/kegg/).

## Screening of candidate *TPS* genes and transcription factors related to D-borneol biosynthesis

Candidate TPS genes were first identified by their KEGG annotation and selected by their length and RPKM expression value. Up-regulated genes were then selected for further analysis. The coding sequences of the candidate *TPS* genes were translated into amino acid sequences, and a phylogenetic tree was reconstructed from the aligned sequences using the neighbor-joining algorithm in MEGA6.0 (*Tamura et al., 2013*) with 1,000 bootstrap replicates. Transcription factors were analyzed using the HMMER method (*Thiriet-Rupert et al., 2016*) and their sequences aligned using the plantTFDB (http://planttfdb.cbi.pku.edu.cn/); they were divided into different families according to their conserved domains.

## Validation of DEGs by qRT-PCR analysis

Quantitative reverse-transcription polymerase chain reaction (qRT-PCR) was conducted using TransStar Tip Green qPCR SuperMix (TransGen Biotech, Beijing, China) and the CFX96 Touch Deep Well platform (Bio-Rad, USA) with a total reaction volume of 20 μL, comprising 1 μL cDNA, 10 μL Tip Green qPCR Supermix, 0.4 μL 10 μM each primer and 8.2 μL nuclease-free water. Nine differentially expressed unigenes related to monoterpenenoid biosynthesis and six candidate transcription factors related to D-borneol accumulation were selected, including CbTPS2 (c122670.graph_c0) and CbTPS3 (c129067.graph_c0), CbGPPS_5 (c117157.graph_c0), CbDXR (c84883.graph_c0), CbDXS_1 (c126058. graph_c0), Cb-DXS_3 (c134634.graph_c0), CbHDS_1 (c134509.graph_c0), CbHDR (c121089.graph_c0), CbHMGR_1 (c133256.graph_c0), CbAP18/ERF-1 (c130684.graph_c1), CbbHLH-1 (c117041.graph_c0), CbbHLH-2 (c127743.graph_c0), CbMYB-1 (c120253.graph_c0), CbMYB-2 (c123637.graph_c0), CbWRKY-1 (c122191.graph_c0). The β-tubulin gene (c121493.graph_c0) was used as a reference. Primer premier 5 software (Premier Biosoft Intl., CA USA) was used to design primers for the selected unigenes; coding sequences and primers are listed in Data S1–S2. The qRT-PCR procedure was as follows: 94 °C for 5 min, 40–45 cycles of 94 °C for 5 s, 60 °C for 15 s and 72 °C for 10 s, followed by a dissociation stage. Expression levels were evaluated using the $2^{-\triangle\triangle Ct}$ method, and the Ct values for all genes were normalized to the Ct value of β-tubulin.

# RESULTS

## D-borneol quantitative determination in three chemotypes of *C. burmanni*

In previous work, we collected three different chemotypes (HBT, LBT, BFT) of *C. burmannii* exhibiting similar morphology (Figs. 2A–2C). These chemotypes possessed high, low and absent levels of D-borneol, respectively. Here, we used a GC/MS method to analyze the

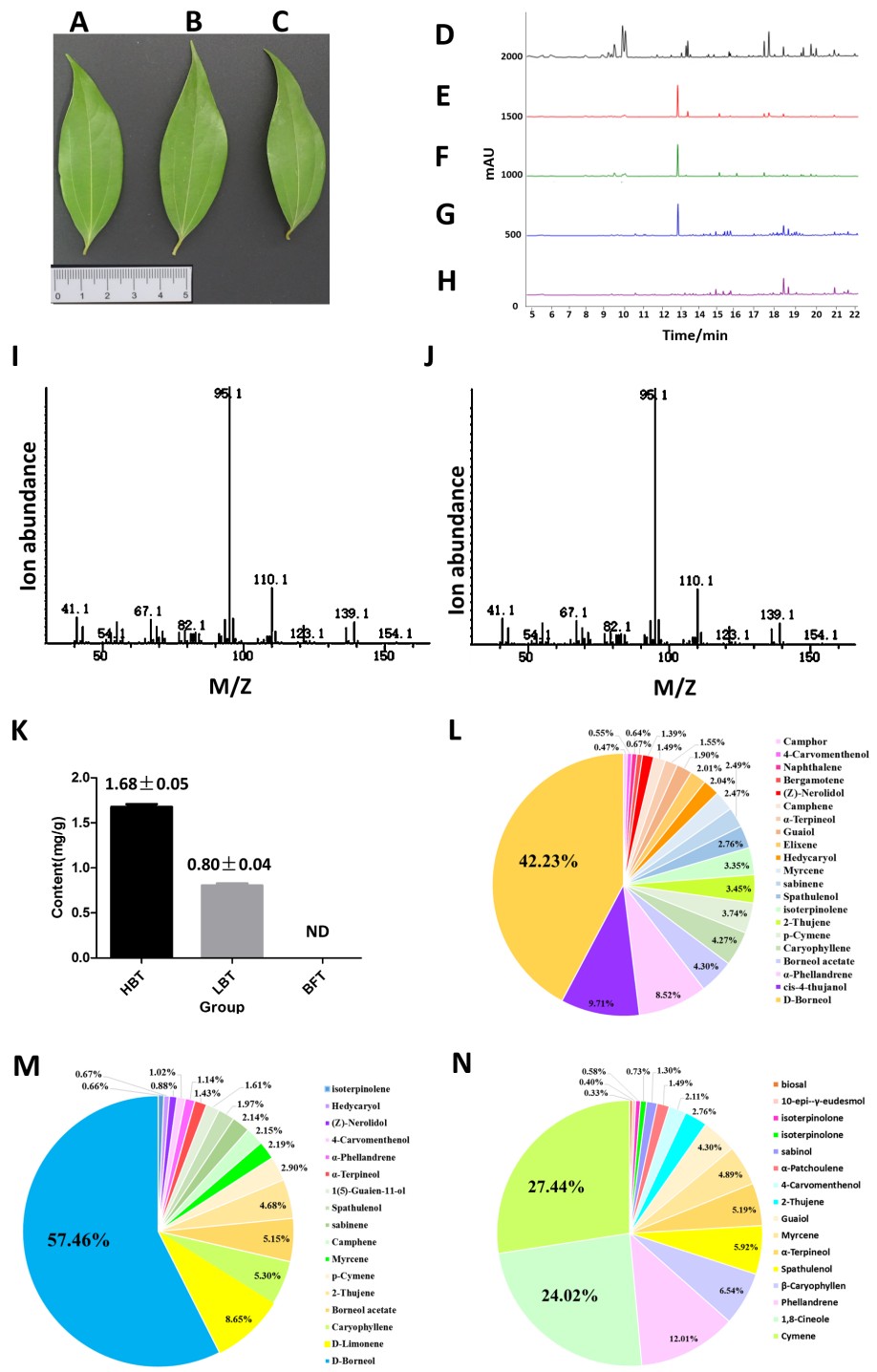

**Figure 2** **Comparison of the content of D-borneol in different chemotype of *C. burmannii*.** (A–C) Leaf morphology of three different chemotypes of *C. burmannii*, while (A) for HBT, (B) for LBT, (C) for BFT; (D–H) GC/MS peak of the *C. burmannii* with different D-borneol content. (D) BFT; (E) LBT; (F) HBT; (G) D-borneol; (H) blank (Petroleum ether); (I) Mass spectra of D-borneol from *C. burmannii* sample; (J) Mass spectra of D-borneol standard. (K) the content of D-borneol were quantified by GC/MS method. ND: not detected. (L) Percentage of terpenes in HBT. (M) Percentage of terpenes in LBT. (N) Percentage of terpenes in BFT.

composition of volatile oil in these *C. burmannii* samples (Figs. 2D–2J). A total of 28 compounds were detected in the three chemotype samples, of which nine were common to all three. D-borneol was the most abundant component of the volatile terpenoids in both HBT and LBT groups, accounting for 42.23% and 57.26%, respectively. In the BFT group, the main component was cymene, accounting for 27.44% of the volatile terpenoids (Figs. 2K–2N, Data S3–S4).

## Transcriptome analysis

To identify genes related to the biosynthesis of D-borneol, we performed deep transcriptome sequencing of the three *C. burmanni* chemotypes using an Illumina HiSeq 2000 platform. In total, 21.21 Gb of clean sequence data with GC content of 47.55% was obtained from all samples after filtering and removing adapter sequences from the raw data: 8.40 Gb from the HBT group, 6.39 Gb from the LBT group and 6.42 Gb from the BFT group. Raw data and quality control data statistics are summarized in Table 1. All RNA-Seq raw data has been submitted to CNGBdb with accession number CNP0000810.

For further analysis, high-quality clean reads were assembled using the TRINITY program, resulting in 142,673 transcripts and 100,218 unigenes with N50 of 1,625 bp and 1,128 bp and average length of 935 bp and 713 bp, respectively. The majority of the reads for both transcripts and unigenes were shorter than 1,000 bp in length, accounting for 92% and 88%, respectively. Only 2% of the transcripts and 3% of the unigenes were longer than 2,000 bp in length. All assembled data are summarized in Table 2 and Fig. 3A. Moreover, 80.80% of the clean reads were perfectly mapped to the assembled unigenes by RSEM, indicating that these mapped genes were of high quality and could be used to conduct the subsequent analysis (Table 1). Sequence of the 100,218 unigenes were shown in Data S5.

All resulting 100,218 unigenes were used for BLAST searches against the GO, KEGG, NR, Pfam and Swiss-Prot databases for annotation, resulting in 28,798, 15,376, 44,495, 28,107 and 24,238 annotated unigenes for the above database, respectively (Fig. 3B, Data S6). In total, 45,485 unigenes were annotated to at least one database, and 9,730 unigenes shared annotation in all five databases (Fig. 3C). The species distribution of the annotated unigenes is shown in Fig. 3D.

## Identification of DEGs

A total of 8,860 DEGs were identified among the comparison groups. The expression patterns of these DEGs could be classified into four clusters (Figs. 3E–3F, Data S7). Cluster 1 consisted of 2,600 genes significantly down-regulated in the LBT group in comparison with both the HBT and BFT groups. Cluster 2 exhibited the opposite expression pattern to that of cluster 1, containing 3,721 unigenes that were particularly up-regulated in the LBT group in comparison to both the HBT and BFT groups. Cluster 3 contained 2,539 unigenes significantly up-regulated in the BFT group in comparison to both the HBT and LBT groups. Ten genes significantly down-regulated in the BFT group in comparison to both the HBT and LBT groups formed cluster 4. We used a Wayne diagram (Fig. 3G) to show the number of DEGs between each group. There were 3,140 up-regulated unigenes and 3,643 down-regulated unigenes in the HBT vs LBT comparison. In the HBT vs BFT

**Table 1** Ttranscriptome sequencing statistics of *C. burmanni* with different borneol content.

| Sample | Raw reads | Clean reads | Clean bases | Q20(%)[a] | Q30(%)[b] | GC content (%) | Mapped ratio |
|---|---|---|---|---|---|---|---|
| HBT | 28,097,241 | 28,097,241 | 8.40 G | 98.02 | 94.61 | 47.67 | 81.34% |
| LBT | 21,355,948 | 21,355,948 | 6.39 G | 98.04 | 94.37 | 47.36 | 80.36% |
| BFT | 22,169,937 | 22,169,937 | 6.42 G | 98.08 | 94.72 | 47.64 | 80.72% |
| Summary | 71,623,126 | 71,623,126 | 21.21 G | – | – | 47.55 | – |

**Notes.**
[a]Q20 indicates the percentage of bases with a Phred value >20.
[b]Q30 indicates the percentage of bases with a Phred value >30.

**Table 2** Length distribution of transcripts and unigenes of the assembly transcritome.

| Item | Transceript | Uinigene |
|---|---|---|
| Total Number | 142,673 | 100,218 |
| Total Length(bp) | 133,416,440 | 71,533,702 |
| N50 Length(bp) | 1,625 | 1,182 |
| Mean Length(bp) | 935 | 713 |

comparison, we identified 1,303 up-regulated unigenes and 924 down-regulated unigenes. In the LBT vs BFT comparison, 3,902 unigenes were up-regulated while 2,357 unigenes were down-regulated (Fig. 3H).

To gain a deep understanding of the biological functions of these 8,860 DEGs, we performed KEGG pathway enrichment analysis. We used dot plots to show the top 20 most enriched KEGG pathway categories with *p* value <0.05, which indicated that the top three most enriched KEGG pathways were "cutin, suberin and wax biosynthesis," "vancomycin resistance" and "fatty acid degradation" (Fig. 4A, Data S8). The top 10 most enriched GO terms under the three general sections Biological Process (BP), Cellular Component (CC) and Molecular Function (MF) are summarized in Fig. 4B and Data S9. The most enriched GO term in the MF category was protein serine/threonine kinase activity. Within the CC category, the most abundant GO term was chloroplast. In the BP category, the most associated term was protein phosphorylation. In addition, terpene synthase activity was also significantly enriched.

## Mining of candidate genes related to D-borneol biosynthesis

Monoterpenes derived from plants are biosynthesized in the plastid through the MEP pathway. We detected 30 unigenes involved in the MEP pathway, but only three of them exhibited significantly different expression levels among the three comparison groups. Most of the unigenes (11/30), including one DEG, were annotated as *DXS* genes, which encode the first rate-limiting enzyme in the MEP pathway. Relatively higher expression of the *DXS* gene in the HBT group compared to that in the other two groups might contribute to the differences in D-borneol accumulation in *C. burmannii*. Six unigenes were annotated as encoding GPPS, responsible for GPP formation. It is well known that GPPSs exist as homomeric and heteromeric structures in plants (*Burke, Klettke & Wise, 2004*; *Rai et al., 2013*; *Wang & Dixon, 2009*; *Yin et al., 2017*) and that plants contain multiple GGPPS or GGPPS-related enzymes, some functioning as large or small subunits (LSU/SSU)

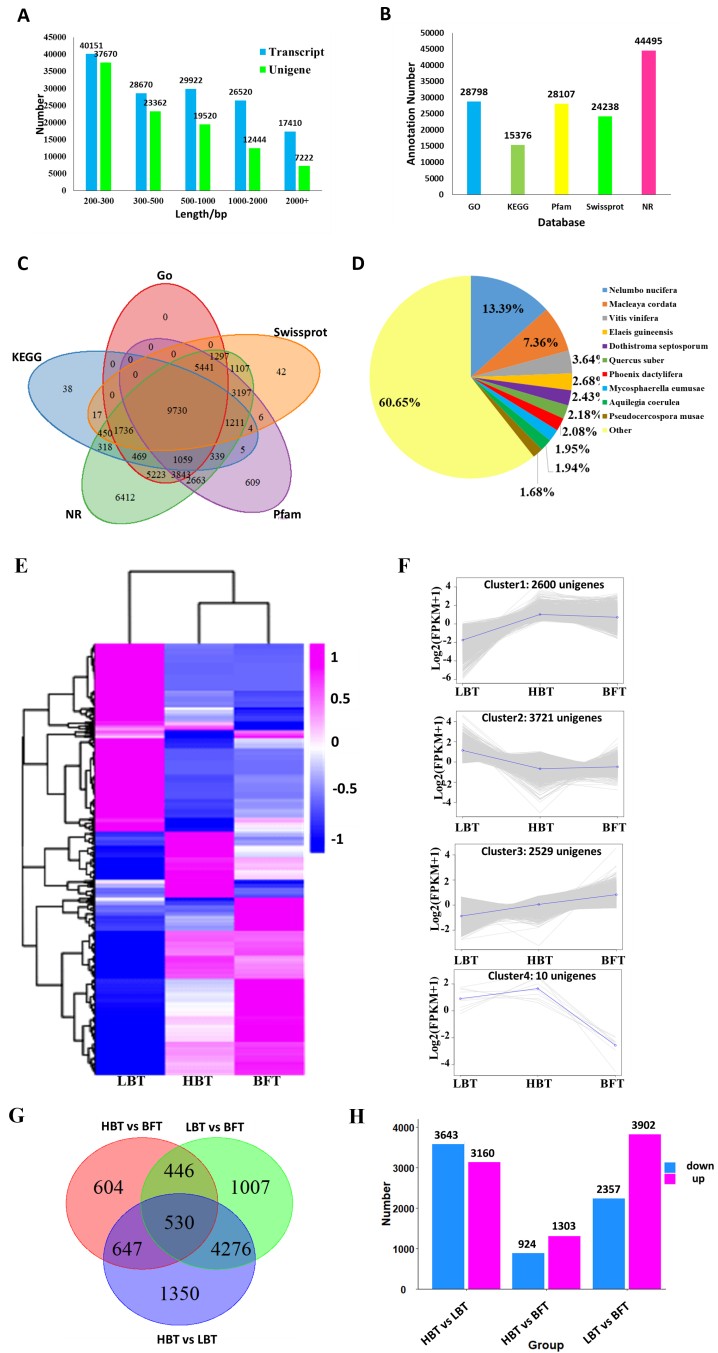

**Figure 3   Transcriptome assembly, annotation results and identification of the DEGs among *C. burmanni* with different borneol content.** (A) The length distribution of unigenes and transcripts of *C. burmanni*. (B) The annotation of unigenes based different databases. (C) Venn diagram of the distribution of annotation information from different public databases. (D) The species distribution of the annotated unigenes. (E) Hierarchical clustering of DEGs among *C. burmanni* with different borneol content. (F) The DEGs were clustered into four clusters by K-means clustering, based on the Pearson correlation distances. (G) Venn-diagram of significantly different DEGs. (H) Number of up- and down-regulated DEGs.

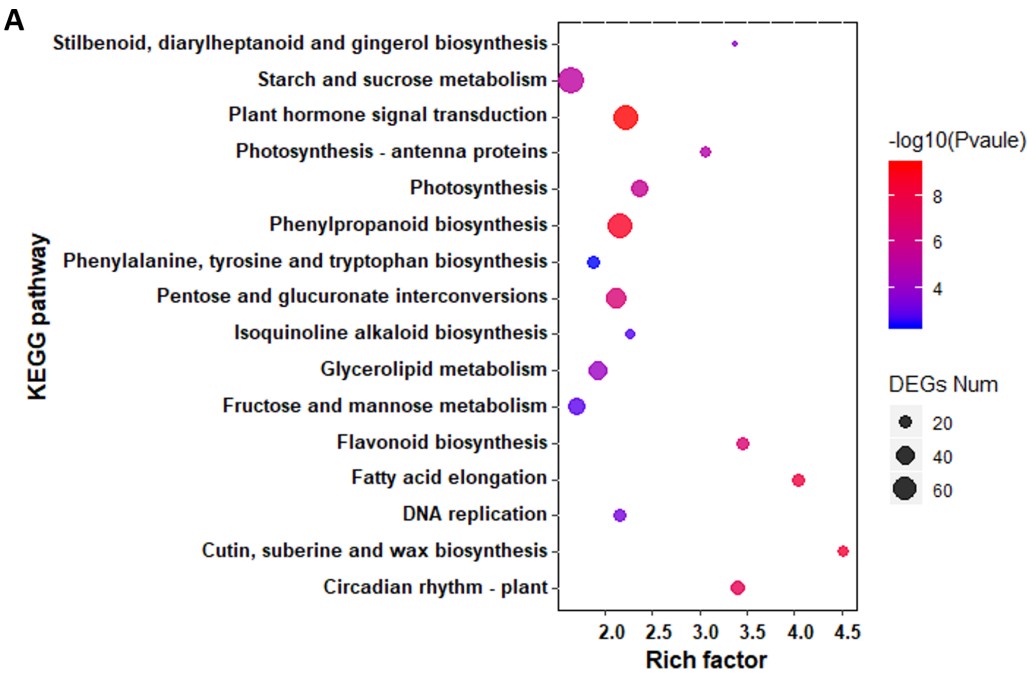

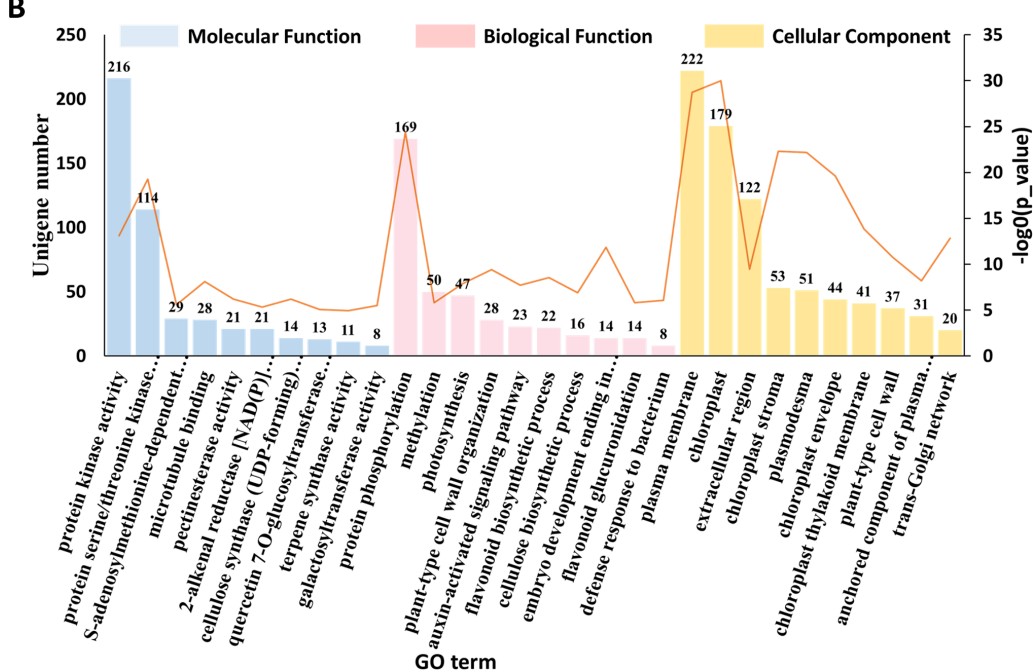

**Figure 4** Functional enrichment analysis of the DEGs among *C. burmanni* with different borneol content. (A) KEGG enrichment analysis of DEGs in the three comparisons. (B) GO enrichment analysis of DEGs in the three comparisons.

of heteromeric GPPS (*Tholl et al., 2004*; *Wang & Dixon, 2009*). For example, there are 12 GGPPS paralogs in the Arabidopsis genome (*Beck et al., 2013*; *Lange & Ghassemian, 2003*); however, AtGPPS11 and AtGGPPS12 were characterized as LSU and SSU, with

their interaction involved in monoterpene biosynthesis in Arabidopsis flowers (*Chen, Fan & Wang, 2015*). To confirm whether the *C. burmannii* GPPSs formed homomeric or heteromeric structures, we conducted a phylogenetic analysis of seven KEGG-annotated CbGPPS/GGPPS proteins from *C. burmannii* and the amino acid sequences of other characterized GPPS/GGPPS proteins. We identified six GPPSs, including two putative homomeric GPPSs, three putative GPPS.SSUs and a putative GPPS.LSU. Only one unigene might encode a GGPPS (Fig. S1). It has been reported that monoterpene biosynthesis in gymnosperms is mainly regulated by homomeric GPPS and SSUs of heteromeric GPPS (*Schmidt et al., 2010*; *Tholl et al., 2004*). The 2 putative homomeric GPPSs, 3 putative GPPS.SSUs identified in this study may provide a possible explanation for the monoterpene richness of *C. burmanni.*

There were 40 unigenes annotated as *TPS* by KEGG, among which 11 were associated with the sesquiterpenoid and triterpenoid biosynthesis pathway (ko00904), eight were monoterpenoid synthase unigenes (ko00902) and 21 encoded diterpenoid synthase (ko00909). Since D-borneol is a monoterpene, the eight putative monoterpenoid synthase genes were selected for further analysis. The expression patterns of these eight unigenes encoding monoterpenoid synthase, along with those of genes related to the MEP metabolic pathway (Ko00900), were normalized and shown in Figs. 5A–5B and Data S10. Considering the relatively high accumulation of D-borneol in the HBT group compared to the other two groups, we rationalized that metabolism-related genes might exhibit more activity in the HBT group than in the other two groups. Inspiringly, two unigenes encoding monoterpenoid synthase (designated *CbTPS2* and *CbTPS3*) were significantly up-regulated in the HBT group compared with the LBT and BFT groups (Data S11). Furthermore, both unigenes had complete open reading frames. Sequence comparison using BLAST indicated that *CbTPS2* (c122670.graph_c0) exhibited highest similarity [53.27% amino acid (aa) sequence identity] to an (E)-beta-ocimene/myrcene synthase (ADR74206.1) from *Vitis vinifera*, while *CbTPS3* (c129067.graph_c0) presented similarity in aa sequence with a S-(+)-linalool synthase (RVW86370.1) from *Vitis vinifera* (*Martin et al., 2010*).

These two candidate *TPS* genes were subjected to a phylogenetic analysis, along with other *TPS* genes that have been functionally characterized. Similar to other well-characterized *BPPS* genes, *CbTPS2* fell within the TPS-b subfamily (Fig. S2). Furthermore, sequence alignment showed that *CbTPS2* had conserved mono-TPS domains, including RRX8W, RXR, DDXXD and (N,D) D (L,I,V) X (S,T) XXE (NSE/DTE) (Fig. S3). To find out whether the candidate unigenes differed between each chemotype, the transcriptomes of the three groups were assembled separately, and the sequences were compared. The assembled sequences of each chemotypes were shown in Data S6–S14. The amino acid sequences of CbTPS2 in the three different *C. burmannii* chemotypes differed only by two amino acids, and these sites were not located within the conserved domain, suggesting that these differences may not affect the function of these proteins (Fig. S4). However, it is interesting that the complete sequence of CbTPS3 could only be found in the HBT and LBT groups, whereas only a small fragment was found in the BFT group (Fig. S5). This may be due to poor assembly of the BFT transcriptome, or perhaps *CbTPS3* is a pseudogene in BFT.
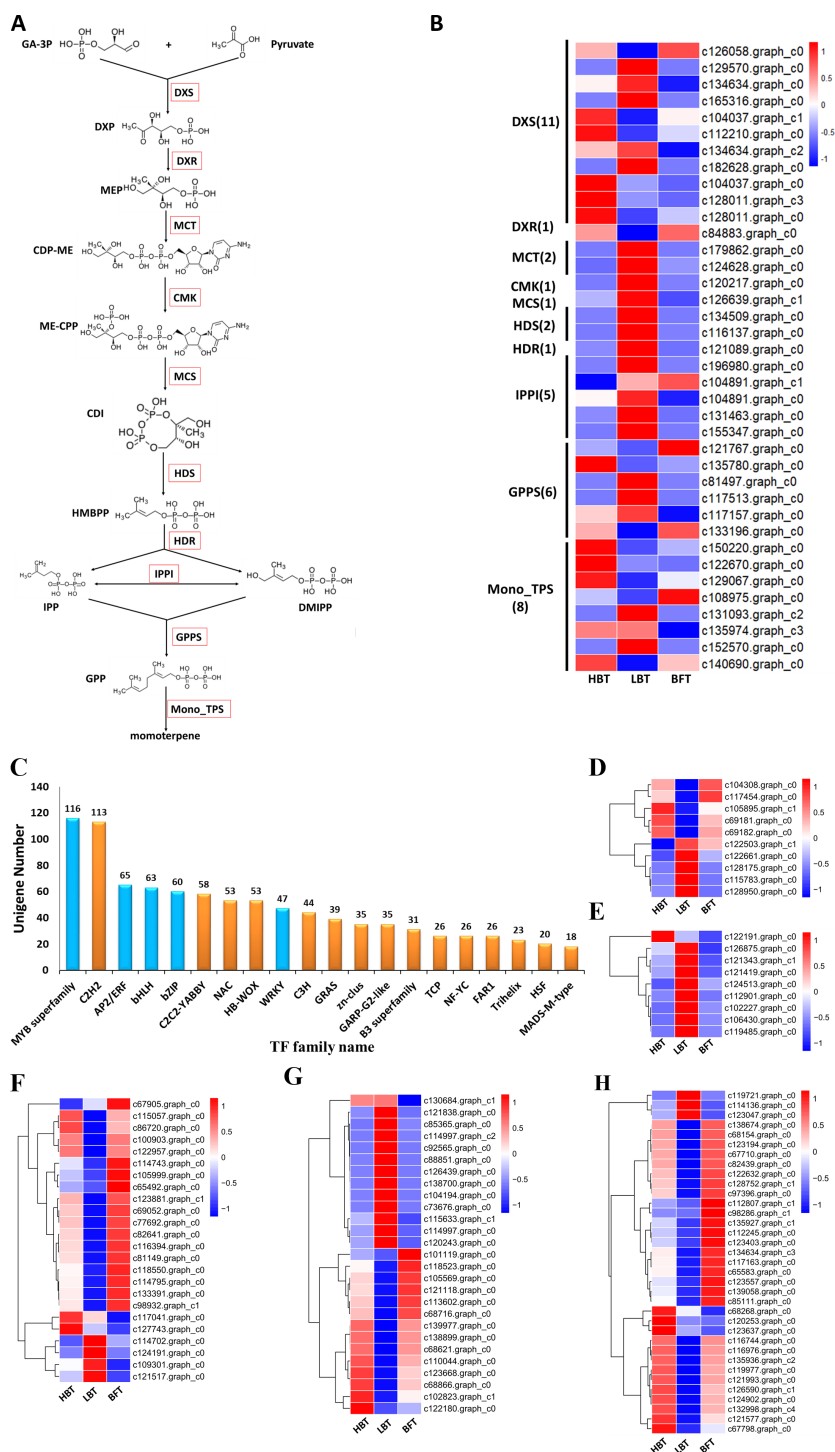

**Figure 5  Genes related to the monoterpenoid biosynthesis pathways in *C. burmanni* and Candidate transcriptional factors analysis.** (A) Brief schematic of monoterpenoid biosynthesis pathway which was modified from KEGG databases. The red box represents (continued on next page...)

**Figure 5 (…continued)**

the enzyme. DXS: 1-Deoxy-D-xylulose-5-phosphate synthase. DXR: 1-Deoxy-D-xylulose 5-phosphate reductoisome-rase. MCT: 2-C-methyl-D-erythritol 4-phosphate cytidylyltransferase. CMK: 4-(Cytidine-5-dipho-spho)-2-C-methyl-D-ery-thritol kinase. MDS: 2-C-methyl -D-erythritol 2,4-cyclodiphosphate synthase. HDS: 4-Hydroxy-3-methylbut-2 -enyl-diphosphate synthase. HDR: 4-Hydroxy-3-meth-ylbut-2-enyl diphosphate reductase. IPPI: isopentenyl diphosphate isomerase. GPPS: geranyl diphosphate syn-thase. DXP: 1-Deoxy-D-xylulose 5-phosphate. MEP: 2-C-Methyl-D-erythritol 4-phosphate. CDP-ME: 4-(Cytidine 5'-diphospho)-2-C-methyl-D-erythritol. ME-CPP: 2-Phospho-4-(cytidine 5'-diphospho)-2-C-methyl-D-erythritol. CDI: 2-C-Methyl-D-erythritol2,4-cyclodiphosphate. HMBPP: 1-Hydroxy-2-methyl-2-butenyl 4-diphosphate. (B) The expression level of the unigenes and the results were shown as a heatmap. (C) The top 20 TF families with the largest number of genes. (D) Expression level of the candi-date BZIP transcription factors. (E) Expression level of the WRKY transcriptional factors. (F) Expression level of the BHLH transcriptional factors. (G) Expression level of the ERF transcriptional factors. (H) Ex-pression level of the MYB transcriptional factors.

We speculated that *CbTPS2* and *CbTPS3* are probably candidate genes responsible for the biosynthesis of D-borneol in *C. burmannii.*

## Mining of candidate transcription factors related to D-borneol accumulation

Transcription factors play an important role in regulating the biosynthesis and accumulation of secondary metabolites (*Hong et al., 2012*). In our transcriptome analysis, we identified 1,108 transcription factors, which could be assigned to 47 families (Data S15). Several TF families, including MYB, basic helix-loop-helix (bHLH), WRKY, basic region/leucine zipper motif (bZIP) and APETALA2/ethylene-response factor (AP2/ERF) families, function in regulating the structural genes involved in monoterpenoid biosynthesis (*Su et al., 2019*). These five TF families cover most of the TFs identified in this study (Fig. 5C). We identified 116 MYB, 65 AP2/ERF, 63 BHLH, 60 bZIP and 47 WRKY candidate genes, among which 35 MYB, 27 AP2/ERF, 24 BHLH, 10 bZIP and 9 WRKY genes were classified as DEGs (Figs. 5D–5H, Data S16). Notably, one WRKY (c122191.graph.c0), two BHLH (c117041.graph.c0, c127743.graph.c0), one AP2/ERF (c130684.graph.c1) and three MYB candidate genes (c68268.graph.c0, c120253. graph.c0, c123637.graph.c0) were significantly up-regulated in both the HBT and LBT groups compared to the BFT group. The different transcription factor expression patterns might be responsible for regulating monoterpenoid biosynthesis. Moreover, the up-regulated transcription factors in the HBT and LBT groups compared to the BFT group provide us with a candidate TF pool related to the biosynthesis and accumulation of D-borneol.

## Validation of DEGs using Quantitative real-time PCR (qRT-PCR)

To verify the accuracy of RNA-Seq data, we performed expression level analysis on nine unigenes related to the terpenoid biosynthesis pathway and six candidate transcription factors related to D-borneol accumulation using the qRT-PCR method. As shown in Fig. 6, their expression patterns were similar to those revealed from RNA-Seq data. Thus, these results help confirm the reliability of the transcriptomic analysis.

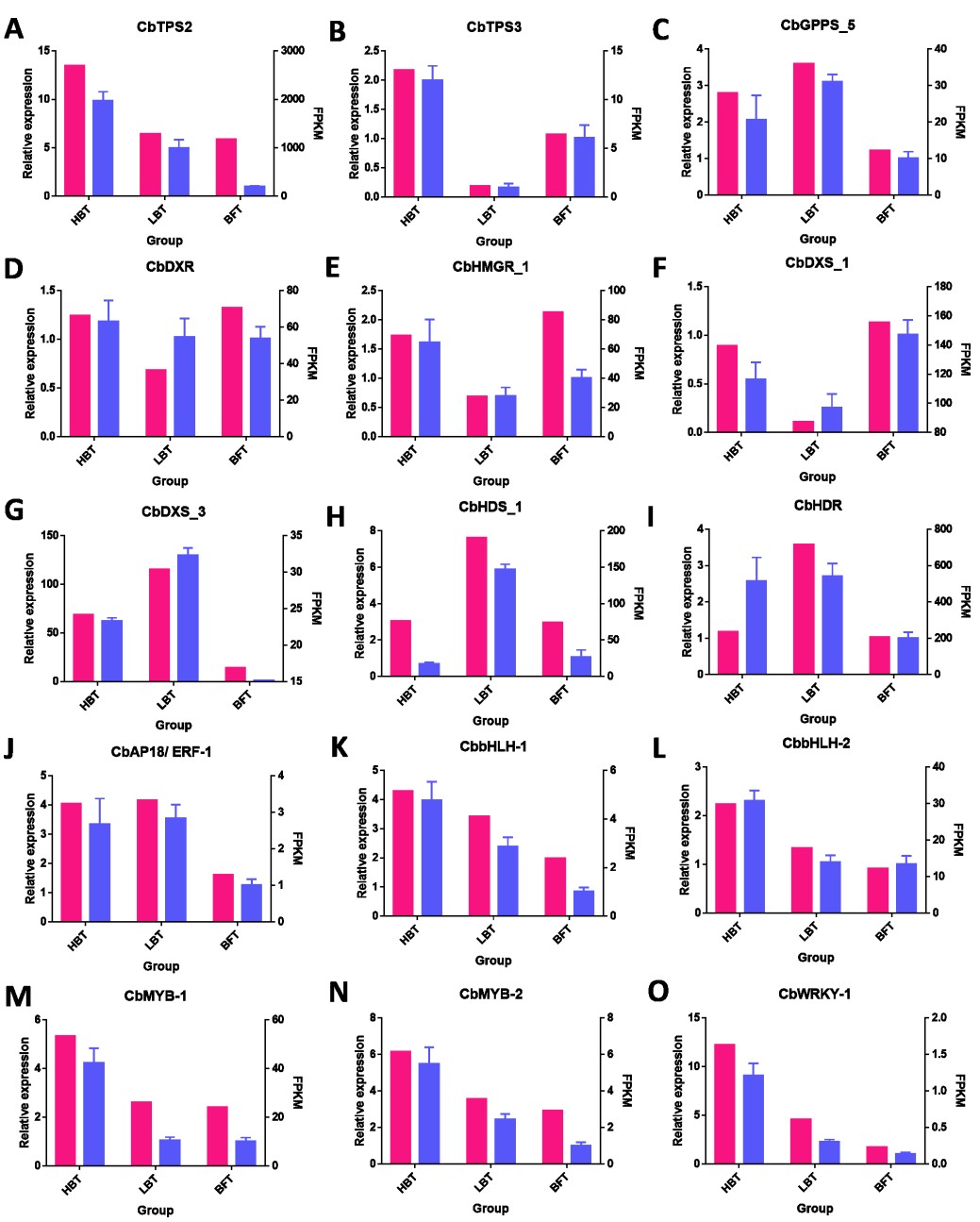

**Figure 6  qRT-PCR validation of genes related to the biosynthesis of terpene from the RNA-seq data.**
(A–O) The FPKM values and the qRT-PCR for the 15 genes selected for this study. The FPKM data of the unigenes were shown in pink bars, while the blue bars indicated the qPCR results. The left Y-axis represents the relative expression calculated by qRCP, and the right Y-axis represents the FPKM value of the RNA sequencing data.

## DISCUSSION

D-Borneol is an effective topical pain reliever for humans, proven in clinical studies (*Wang et al., 2017*). It is therefore attracting more and more attention from researchers. In China,

current D-borneol production is from fresh leaves of *C. burmannii* by water distillation. However, balancing the supply and demand of D-borneol is a challenge due to the long plant growth period, unsatisfactorily low yield and the huge consumption of labor and energy. Authenticating high-efficiency enzymes associated with the biosynthesis of D-borneol will benefit heterologous recombination, which could help significantly in solving the supply problem. However, deficiency of *C. burmannii* genomic and transcriptomic information increases the difficulty of identifying core enzymes associated with D-borneol biosynthesis. Our transcriptome analysis of *C. burmannii* may help to uncover genes related to the biosynthesis of D-borneol through bioinformatics analysis.

It is a common phenomenon that noteworthy differences in chemical composition exist between individuals of the same species in the genus *Cinnamomum* (*Guo et al., 2017*; *Singh et al., 2012*; *Teles et al., 2019*). For example, there are at least five chemotypes in *C. camphora*, including the borneol type, linalool type, camphor type, nerolidol type and 1,8-cineole type (*Chen et al., 2018*). *Cinnamomum tenuipilum* harbors six chemotypes, including the methyleugenol type, citral type, linalool type, geraniol type, camphor type and farnesol type (*Cheng et al., 1991*). In *C. burmannii*, we also found great differences in the content of D-borneol in different individuals, despite it being reported to be a D-borneol rich species. The molecular mechanism of chemotype formation in *Cinnamomum* has not yet been elucidated, but studies on other plants such as *Melaleuca alternifolia*, which is rich in terpenes and has different chemotypes, indicates that a difference in TPS may be responsible for the phenomenon of multiple chemotypes within the species (*Bustos-Segura et al., 2017*; *Padovan et al., 2017*).

Transcriptome analysis is currently the most widely used and cost effective way to screen target gene(s) rapidly in functional genomic studies (*Ali et al., 2018*; *Ma et al., 2019*; *Zhou et al., 2019*). In this study, a total of 100,218 unigenes were generated by transtomic sequencing of three chemotypes of *C. burmannii*. In detail, 45,485 unigenes (45.38%) succeeded in functional annotation against several public databases. However, the annotation rate of unigene assembled in this study appear to be quite low, compared to other species, which may be due to the variable splicing of transcripts and the deficient of genomic sequences (*Liang et al., 2019*).

In the past few decades, genes involved in the biosynthesis of terpenes have been verified in many plants (*Abbas et al., 2019*; *Anand et al., 2019*; *Vishal et al., 2019*). However, although species within the *Cinnamomum* genus are rich in terpenes, only one gene encoding geraniol synthase has been isolated and functionally characterized from *C. tenuipilum* (*Yang et al., 2005*). Transcriptome study of different chemotypes of *C. camphora* indicated that 67 genes might be involved in terpene biosynthesis, of which three genes (CcTPS14-like1, CcTPS14-like2, CcTPS14-like3) were significantly upregulated in the borneol chemotype relative to the linalool chemotype. These three genes may play an important role in terpenoid accumulation in the borneol-type of *C. camphora* (*Chen et al., 2018*). Transcriptome analysis of three chemotypes of *C. porrectum* identified 52, 49 and 66 candidate unigenes related to terpene biosynthesis in the eucalyptol chemotype and linalool chemotype, the camphor chemotype and eucalyptol chemotype, and the camphor chemotype and linalool chemotype comparison groups, respectively. The terpenoid

synthase genes *CpTPS1*, *CpTPS3*, *CpTPS4*, *CpTPS5* and *CpTPS9* had specific expression or higher expression in one of the chemotypes and may be involved in monoterpene biosynthesis (*Qiu et al., 2019*). The transcriptome and bioinformatics analysis performed in this study aimed to uncover candidate genes related to the biosynthesis of D-borneol. Compared to the publishedCinnamomumtranscriptome, we found more genes related to terpenoid biosynthesis, but fewer genes related to specific terpenoid biosynthesis. We annotated a total of 82 genes as related to terpene biosynthesis, 52 of which participated in the biosynthesis of the terpene backbone and 30 of which were annotated asTPS (Data S17). Only 13 of these 94 genes were DEGs, including oneDXSgene and two genes encoding monoterpene synthase. DXS is regarded as the first rate-limiting enzyme in the MEP pathway, which provides corresponding precursors for monoterpene and diterpene. Of the two monoterpene synthases identified in this study,*CbTPS3* was annotated as a *TPS14-like* gene, which mediates the biosynthesis of linalool as reported in *Arabidopsis thaliana* (*Chen et al., 2003*). Interestingly, although *CbTPS3* was actively expressed, no or little linalool was found in *C. burmannii.* This phenomenon was also observed in the closely related species *C. camphora* (*Chen et al., 2018*). Furthermore, we found that CbTPS3 probably only exists in the borneol chemotype of *C. burmannii.* Therefore, whether *CbTPS3* is involved in the biosynthesis of D-borneol needs further examination. The other monoterpene synthase gene,*CbTPS2*, was clustered into the TPSb subfamily. Nevertheless, functional authentication remains to be verified in a future study, regarding to the low homology among those reported known TPSb-subfamily genes which have the same catalytic function (*Hosoi et al., 2004*; *Landmann et al., 2007*).

Transcription factors activate synergistic expression of multiple genes encoding secondary metabolite synthases, thereby effectively regulating secondary metabolic pathways. Plant terpenoid biosynthesis is mainly regulated by transcription factors, such as bZIP, WRKY, bHLH, AP2/ERF and MYB family members. For example, a PGT-specific *R2R3-MYB* gene, *MsMYB*, from *Mentha spicata* regulates monoterpene production and suppresses the expression of the geranyl diphosphate synthase large subunit (*Reddy et al., 2017*). Transient ectopic expression of *bZIP* TF genes isolated from *Phalaenopsis bellina*, designated *PbbZIP*, induces a 10-fold increase in monoterpenoid production in the scentless orchid (*Chuang et al., 2018*). A jasmonate-responsive bHLH factor participates in modulating monoterpenoid indole alkaloid biosynthesis in *Catharanthus roseus* (*Patra et al., 2018*). Transcription factor *CitERF71* activates the terpene synthase encoding gene *CitTPS16* involved in the biosynthesis of E-geraniol in *Citrus sinensis* Osbeck (*Li et al., 2017*). Recently, a genome-wide investigation of WRKY transcription factors in Osmanthus Fragrans indicated potential roles for OfWRKY139 and OfWRKYs with plant zinc cluster domains in regulating biosynthesis of aromatic compounds in sweet osmanthus (*Ding et al., 2019*); A study on monoterpene synthase promoter related transcription factors in *Lavandula × Intermedia* by yeast one-hybrid assay indicated that multiple transcription factors control monoterpene synthase expression in lavender, including MYB, bZIP, NAC, GeBP and SBP-related proteins (*Sarker, Adal & Mahmoud, 2019*). In the present work, we identified one *WRKY*, two *BHLH*, one *AP2/ERF* and three *MYB* candidate genes, which exhibited the same expression pattern as *CcTPS5* and *CcTPS7*. Based on the idea that genes

co-expressed under multiple conditions tend to be functionally related (*Aoki, Ogata & Shibata, 2007*; *Ihmels, Levy & Barkai, 2004*), elucidating the function of these transcription factors in D-borneol regulation will be the focus of follow-up research.

## CONCLUSION

To the best of our knowledge, this is the first transcriptomic study in *C. burmannii*, aimed at discovering genes participating in the biosynthesis of D-borneol. A total of 100,218 unigenes were assembled from our transcriptome sequencing of *C. burmannii* with three different levels of borneol, among which 8,860 unigenes were defined as DEGs. Two of these DEGs, *CbTPS2* and *CbTPS3*, had relatively high expression levels in the *C. burmannii* chemotype with high D-borneol content. We conclude that these two genes might be directly responsible for the biosynthesis of D-borneol. Furthermore, seven TF genes (one *WRKY*, two *BHLH*, one *AP2/ERF* and three *MYB*) possessing the same expression pattern as *CbTPS2* and *CbTPS3* might play an important role in regulating D-borneol biosynthesis. However, their specific functions need to be elucidated in further studies. The extensive gene assembly obtained from our transcriptomic sequencing provides a pool of candidate genes for analyzing the accumulation mechanism of D-borneol in *C. burmannii*.

### Funding

This research was supported by the National Natural Science Foundation of China (Grant number: 81903741). The funders had no role in study design, data collection and analysis, decision to publish, or preparation of the manuscript.

### Grant Disclosures

The following grant information was disclosed by the authors:
National Natural Science Foundation of China: 81903741.

### Competing Interests

The authors declare there are no competing interests.

### Author Contributions

- Zerui Yang conceived and designed the experiments, performed the experiments, analyzed the data, prepared figures and/or tables, authored or reviewed drafts of the paper, and approved the final draft.
- Wenli An conceived and designed the experiments, performed the experiments, analyzed the data, authored or reviewed drafts of the paper, and approved the final draft.
- Shanshan Liu, Yuying Huang and Chunzhu Xie conceived and designed the experiments, performed the experiments, authored or reviewed drafts of the paper, and approved the final draft.
- Song Huang and Xiasheng Zheng conceived and designed the experiments, analyzed the data, prepared figures and/or tables, authored or reviewed drafts of the paper, and approved the final draft.

## Data Availability

Data is available at CNGBdb: CNP0000810.

## Supplemental Information

Supplemental information for this article can be found online at http://dx.doi.org/10.7717/peerj.9311#supplemental-information.

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
