# Peer review of "Mining of candidate genes involved in the biosynthesis of dextrorotatory borneol in Cinnamomum burmannii by transcriptomic analysis on three chemotypes"

_PeerJ, doi:10.7717/peerj.9311_

## Round 0.1 · original submission · Major Revisions

Three experts in the field evaluated this submission. They all have concerns related to this manuscript. I recommend a major revision in this paper.

Reviewer 1 ·

Basic reporting

The manuscript would benefit from english editing, this is based on the following examples:
Line 34: "When applied orally administration"
Line 66: "prenyl diphosphatesare" should read "prenyl diphosphates are"
Line 69: "under the catalysis" is not the right term
Line 193: "chmotyps"
Line 197: "statistics were concluded"
Line 248: "exhibit more actively"
Line 299: "proved by"
.......

The cited literature is sufficient; however it favors citation of Chinese researchers.

The article structure, figures and tables are largely professional, except for Figure1a I do not think that it is necessary to include a picture of crystalline D-borneol in the article. Figure 2b the chromatogram looks like a screenshot and the line indicating the retention time of D-borneol makes the peaks hard to see. Moreover, from the legend it is not clear if c is a spectrum of D-borneol in a sample or from the NIST-library. Figure 6: the legend speaks about black and grey bars, however the bars are pink and blue.
The raw data is shared, however the link to the transcriptomics data supplied with the review material is not found in the manuscript itself. In line 204 of the results the authors talk about a reference transcriptome for mapping, but in the Material and Methods no reference transcriptome is specified.

The largest drawback of this research is that the core goal to identify a D-borneol synthase is purely hypothetical. While they claim to have identified two terpene synthases that are good candidates, I disagree. First of all, these enzymes were not characterized at all and terpene synthases are notorious for producing multiple products, therefore while these enzymes might produce D-borneol this might be only one terpene among many and not even the major product. This would prevent using this enzyme for biotechnological production of d-borneol without further protein engineering near impossible. The best way to improve the manuscript and support the claim that these enzymes could be used for biotechnological approaches, would be to express the two enzymes in a well-established heterologous host like E. coli and either use a metabolic engineering approach or purify protein for in vitro assays.

Experimental design

The research tries to fill a knowledge gap, however it falls short as stated above in supporting the conclusion that this research and the investigated genes will help in the intended biotechnological approach for the production of d-borneol.

Validity of the findings

Again the conclusions overstate the findings, as the two suggested genes were not characterized and might not be the searched for D-borneol synthase.

Additional comments

The authors identified GPPS and their expression pattern (Figure 5), which enzyme was used for anotation? The previously identified GPPS in Arabidopsis was later identified to produce longer chain prenyl diphosphates (Hsieh FL, Chang TH, Ko TP, Wang AHJ (2011) Structure and mechanism of an Arabidopsis Medium/long-chain-length prenyl pyrophosphate synthase. Plant Physiol 155(3):1079–1090) and instead a heteromeric GGPPS is producing GPP (Chen Q, Fan D, Wang G (2015) Heteromeric Geranyl(geranyl) diphosphate synthase is involved in monoterpene biosynthesis in Arabidopsis Flowers. Mol Plant.).

Reviewer 2 ·

Basic reporting

The manuscript entitled “Mining of candidate genes involved in the biosynthesis of dextrorotatory borneol in Cinnamomum burmanni by transcriptomic analysis on three chemotypes” by Zerui Yang et al. identified three structural genes and seven transcription factors involved in the D-borneol biosynthesis in Cinnamomum burmanni. The manuscript is overall well-structured and focused. I think the major conclusions are well supported. The findings are interesting and will be utilized in the field. I have only minor concerns:

Line 32: Replace “administration” with “administrated”.
Line 193: Spell check for chemotypes.
Please refer Caihui Chen et al. (doi: 10.1186/s12864-018-4941-1) in introduction
Line207: replace blast with BLAST
Line 256: replace “partiality” with “partial”
Figure 5: In figure legend replace factorsa with factors

Experimental design

Experiments are well designed and executed.

Validity of the findings

Findings are novel, interesting and useful for future studies within field.

Additional comments

Manuscript is suitable for publication in PeerJ after minor correction.

Reviewer 3 ·

Basic reporting

The article meets the Journal's standards.

Experimental design

The experimental is well designed.

Validity of the findings

More DEGs should be conformed by the QRT-PCR. For the DEGs that be discussed in the discussion section. For examples, For example, candidate transcription factors related to D-borneol accumulation.

Additional comments

1. What are the means of HBT, LBT and BFT groups? They should be well defined in the M&M section.
2. Line 214-216 should be moved to the M&M section.
3. To date, many transcriptome analyses have been performed in Cinnamomum plants. However, no one has been mentioned in the Introduction.
4. The differences in the genes related to the synthesis of terpenes should be compared with the previous published Cinnamomum transcriptomes. Are there any differences among them? More unigenes have been identified in the present study?
5. The LBT, HBT and BFT are three different chemotypes of C. burmanni, though they exhibit similar morphology. Are the sequences of key genes involved in synthesis of terpenes also similar?
As I saw, three transcriptomes from LBT, HBT and BFT were assembled together. I suggested that three transcriptomes should be assembled separately. Then, the differences in gene sequences should be compared and analysis. Maybe, this is the reason of the difference in borneol content.
6. For the QRT-PCR, more DEGs should be validated. For example, candidate transcription factors related to D-borneol accumulation.
7. For monoterpene biosynthesis, many TFs have been reported to be involved. In the discussion section, the novel insights on the TFs involved in monoterpene biosynthesis should be highlighted.

---

## Round 0.2 · Minor Revisions

The reviewers recommend accepting this submission as it is. On the other hand, the section editor asked for a minor revision as follows:

"The manuscript has a lot to offer; however, the authors are very stingy on letting the audience be aware of the data. The data, meaning raw data, needs to be provided; otherwise it is not supported or substantiated. Annotation need to be paired with the actual unigenes described. There are only 15 unigenes listed in S2. There are no unigenes attached to the KEGG and GO data in S4 and S3. There is a lot of nice figures, but there is no actual data made available to back it up; there should be at least 9730 sequences; at least. The manuscript has a good introduction and discussion of findings, but no data (just pretty pictures). To bring this up to speed will probably require some modifications to the manuscript and data structure. I would place this in a moderate revision category. A better connection between the results and the data that led them to the conclusions is needed."

Please carry out the modifications as recommended by the section editor.

Reviewer 1 ·

Basic reporting

The Manuscript has significantly improved from the previous version. I still think that inlcuding biochemical data would have made this manuscript way stronger in its impact, but it should also not preventing it from being published.

Experimental design

no additional comments

Validity of the findings

no additional comments

Additional comments

In the rebutal you shared SDS-PAGE pictures that show the TPS being in inclusion bodies, the reason for this might be that you used a Vector with N-terminal His-Tag on a truncated plastid protein. In this case mostly c-terminal Tags were used, as the are not as detrimental for folding as an N-terminal Tag.

Reviewer 2 ·

Basic reporting

The manuscript has improved a lot after revision.

Experimental design

no comments

Validity of the findings

Authors have incorporated all my suggestions

Additional comments

I think the manuscript can now be considered for publication in PeerJ.

---

## Round 0.3 · accepted · Accept

The authors have carried out all modifications indicated by the reviewers and also they followed the suggestions made by the section editor. In my view, the manuscript meets the high standards necessary for publication in PeerJ. I recommend accepting this paper as it is.